# Discovery of a New Xanthone against Glioma: Synthesis and Development of (Pro)liposome Formulations

**DOI:** 10.3390/molecules24030409

**Published:** 2019-01-23

**Authors:** Ana Alves, Marta Correia-da-Silva, Claúdia Nunes, João Campos, Emília Sousa, Patrícia M.A. Silva, Hassan Bousbaa, Francisca Rodrigues, Domingos Ferreira, Paulo C. Costa, Madalena Pinto

**Affiliations:** 1UCIBIO, REQUIMTE, Laboratory of Pharmaceutical Technology, Faculty of Pharmacy, University of Porto, Rua de Jorge Viterbo Ferreira, 228, 4050-313 Porto, Portugal; anadaniela92@hotmail.com (A.A.); jcampos@ff.up.pt (J.C.); domingos@ff.up.pt (D.F.); 2Laboratory of Organic and Pharmaceutical Chemistry, Department of Chemical Sciences, Faculty of Pharmacy, University of Porto, Rua Jorge Viterbo Ferreira, 228, 4050-313 Porto, Portugal; m_correiadasilva@ff.up.pt (M.C.-d.-S.); madalena@ff.up.pt (M.P.); 3Interdisciplinary Centre of Marine and Environmental Research (CIIMAR), University of Porto, Terminal de Cruzeiros do Porto de Leixões Avenida General Norton de Matos P 4450-208 Matosinhos, Portugal; hassan.bousbaa@iucs.cespu.pt; 4LAQV, REQUIMTE, Departamento de Ciências Químicas, Faculdade de Farmácia, Universidade do Porto, Rua de Jorge Viterbo Ferreira, 228, 4050-313 Porto, Portugall; cdnunes@ff.up.pt; 5CESPU, Institute of Research and Advanced Training in Health Sciences and Technologies (IINFACTS), Rua Central de Gandra, 1317, 4585-116 Gandra, Portugal; patricia.silva@cespu.pt; 6REQUIMTE/LAQV, Instituto Superior de Engenharia do Porto, Instituto Politécnico do Porto, Portugal; franciscapintolisboa@gmail.com

**Keywords:** xanthone, acetylated, glycosylation, synthesis, glioblastoma, tumor cell lines, nanotechnology, liposomes, proliposomes

## Abstract

Following our previous work on the antitumor activity of acetylated flavonosides, a new acetylated xanthonoside, 3,6-bis(2,3,4,6-tetra-*O*-acetyl-β-glucopyranosyl)xanthone (**2**), was synthesized and discovered as a potent inhibitor of tumor cell growth. The synthesis involved the glycosylation of 3,6-di-hydroxyxanthone (**1**) with acetobromo-α-d-glucose. Glycosylation with silver carbonate decreased the amount of glucose donor needed, comparative to the biphasic glycosylation. Xanthone **2** showed a potent anti-growth activity, with GI_50_ < 1 μM, in human cell lines of breast, lung, and glioblastoma cancers. Current treatment for invasive brain glioma is still inadequate and new agents against glioblastoma with high brain permeability are urgently needed. To overcome these issues, xanthone **2** was encapsulated in a liposome. To increase the well-known low stability of these drug carriers, a proliposome formulation was developed using the spray drying method. Both formulations were characterized and compared regarding three months stability and in vitro anti-growth activity. While the proliposome formulation showed significantly higher stability, it was at the expense of losing its biocompatibility as a drug carrier in higher concentrations. More importantly, the new xanthone **2** was still able to inhibit the growth of glioblastoma cells after liposome formulation.

## 1. Introduction

Current treatment for invasive brain glioma is still inadequate, and prognosis upon diagnosis tends to be very poor. Several factors contribute to these limitations, such as the highly invasive, non-localized and diffuse characteristics of the tumors, and the difficulty of local drug activity [1,2]. Conventional surgical methods and/or radiotherapy alone cannot eliminate cancer cells from the brain, and the relapse is, most of the time, inevitable [3]. Temozolomide, an alkylating agent, remains the standard-of-care in glioma chemotherapy. However, chemotherapy for gliomas is difficult due to two major obstacles: The blood-brain barrier (BBB) and the heterogeneity of the brain cancer [4]. Very recently, substitution of the amide group of temozolomide for a methylketone increased brain permeability with 69% entering the central nervous system compared with just 8% for temozolomide and produced a more effective compound when tested in mice [5]. Temozolomide was also recently encapsulated in liposomes for brain tumor treatment [6].

Natural glycosides of flavonoids and xanthones demonstrate several biological activities in which the glycosidic moiety showed an important role [7,8]. In our group, an acetylated flavonoid glycoside obtained by synthesis was recently shown to have a potent inhibitory effect on glioma cells growth [9]. Association of acetyl groups with glioma cell lines growth was also reported in the natural *O*-acetylated ganglioside GD1b neurostatin, present in the mammalian brain, which showed strong inhibition of astroblast and astrocytoma division [10]. Furthermore, acylation is expected to increase the penetration through the cell membrane [11,12]. On the other hand, the rigid heteroaromatic tricyclic core of xanthones was considered as a privileged structure and several representatives are in clinical research for cancer treatment [13]. Based on these considerations, we hypothesized that acetylated xanthone glycosides could render potent glioma cell growth inhibitors and incorporated in proliposomes could allow blood-brain barrier penetrance while avoiding hydrolysis by esterases.

Liposomes, reported firstly in 1965 by Bangham et al., are microscopic vesicles of one or more concentric lipid bilayers separated internally and externally by an aqueous medium [14]. Liposomes may incorporate hydrophilic and/or lipophilic substance, where they can: (i) Stand in the aqueous phase, (ii) be inserted in lipophilic phase, or (iii) be adsorbed in the membrane surface [15,16]. These vesicles are composed of phospholipids and can be of synthetic or natural origin [17]. The most used lipids in the liposome formulations are those who have a cylindrical shape as phosphatidylcholines, phosphatidylserine, phosphatidylglycerol, and sphingomyelin, which lead to the formation of stable bilayer in aqueous solution [18]. In this case, the liposomes can be better absorbed due to the weakening of the BBB near the glioma (passive targeted based on the EPR effect) [19]. Moreover, liposomes proved to be a valuable approach to transport lipophilic xanthones across BBB, both in in vitro and in vivo models [20].

The proliposomes, developed by Payne in 1986 [21], are a new generation of drug delivery system (DDS), having several advantages over conventional liposomes, such as the stability improvement or the easy sterilization on a large scale [22,23]. These systems are composed of a carrier, which is a water-soluble porous powder, where phospholipids, drugs, and other components are molecularly dispersed. They can be defined as a dispersed system of free flowing particles that can immediately form a liposomal suspension in contact with water. The fact that the drug is encapsulated increases its stability and produces a sustained release effect at the administration site [24,25,26]. The development of proliposomes has been used for administration by oral (e.g., silymarin and indomethacin), pulmonary (e.g., levofloxacin, isoniazid, and pyrazinamide), and parenteral (indomethacin, carboplatin, polymyxin E, docetaxel, and glycyrrhetinic acid) routes [21,27,28,29,30,31]. The encapsulation of anticancer drugs in liposomes after hydration of proliposomes seems to alter the pharmacokinetic and tissue distribution profiles and increase the therapeutic effect [32]. Some anticancer drugs have been encapsulated using proliposomes (doxorubicin, carboplatin, and docetaxel) [33].

In our group, both natural and synthetic antitumor xanthone derivatives were already encapsulated in micro and nanoparticles containing biocompatible and biodegradable polymers to improve their antitumor effect [34,35]. Chen et al. prepared transferrin modified liposomes of the polyphenolic xanthone α-mangostin, and demonstrated that this polar xanthone was able to penetrate the brain in the form of intact liposome [20]. In the present study, a symmetric xanthone with two acetylated glycosidic portions, obtained by synthesis, was disclosed as a potent glioma cell growth inhibitor and liposomes and proliposomes were developed to assure penetration through BBB.

## 2. Results and Discussion

### 2.1. Synthesis and Structure Elucidation

3,6-Bis(2,3,4,6-tetra-*O*-acetyl-β-glucopyranosyl)xanthone (**2**, XGAC) was obtained by the reaction of 3,6-dihydroxyxanthone (**1**) with α-acetobromoglucose as the glucose donor (Scheme 1), using two methods: Michael and Koenigs-Knorr methods. Firstly, glycosylation was performed according to previously described procedures [36] by biphasic Michael reaction—phase transfer catalyzed (PTC) coupling reaction. In the PTC method, a two-phase solvent of chloroform/aqueous potassium carbonate and a phase-transfer catalyst was employed. However, 12 g of the glucose donor (7 equiv/OH) were required for the glycosylation of 800 mg of xanthone **1** yielding 20% of the pure glycosylated xanthone **2**, which made this reaction very expensive and ineffective for scale up. Moreover, during liquid/liquid extraction this glycosylated xanthone **2** insolubilized in both phases, making the workup process tedious. In an attempt to increase the efficiency of xanthone **1** glycosylation, the Koenigs-Knorr method was applied. Koenigs-Knorr reaction is one of the most useful reactions in obtaining *O*-glycosides. Usually, silver salts and drying agents are used. Using a mixture of dry acetonitrile and acetone it was possible to dissolve 3 g of xanthone **1,** and in the presence of silver carbonate, only 2 equiv/OH of α-acetobromoglucose were needed, although with a lower yield. Nevertheless, glycosylation with silver carbonate was more feasible for scaling-up and with a shorter reaction time than the previously applied PTC glycosylation. In both processes, a bimolecular process operates, resulting in inversion of stereochemistry at the anomeric center.

For biological activity comparison purposes, 3,6-bis(*O*-β-d-glucopyranosyl)xanthone (**3**) was obtained according to previously described procedures [36] using a Zemplén deacetylation with sodium methoxide solution (0.5 M in methanol) at room temperature (Scheme 1). Zemplén deacetylation was selected because of the use of catalytic amounts of base (1 equiv/acetyl group of sodium methoxide), short reaction times, and excellent yields. In only 1 h, the reaction was complete, and neutralization using an ion exchange resin (DOWEX H^+^ form) was carried out to afford compound **3** in 92% yield.

The structure of compound **2** was established for the first time, using infrared (IR), nuclear magnetic resonance (NMR, Appendix A), and high-resolution mass spectrometry (HRMS) techniques. The IR spectrum of compound **2** showed a strong band at 1750 cm^−1^ typical of C=O ester vibration. ^1^H- and ^13^C-NMR spectra of compound **2** indicated the presence of a symmetric xanthone with two glycosidic moieties. The inversion of stereochemistry at the anomeric center of the glucose donor was confirmed by the coupling constant of 7 Hz of the doublet at δH 5.24 ppm (H-1′), indicating a β-configuration. The position of the sugar moiety in compound **2** was evidenced by the correlation found in the heteronuclear multiple bond correlation (HMBC) spectrum between δ_H_ 5.36–5.17 ppm and δ_C_ 161.4 ppm.

### 2.2. Human Tumor Cell Lines Growth Inhibitory Activity of Xanthone *(**2**)*

Acetylated structures are often unstable in cells and may be metabolized into the corresponding alcohols/phenols. Therefore, the non-acetylated compound **3** was synthesized to allow an understanding of the role of the acetyl group on the cell growth inhibitory activity. The starting material **1** was previously shown to have a weak cell growth inhibitory activity in breast, lung, and glioma cell lines [37]. Therefore, acetylated xanthonoside **2** and its derivative **3** were first evaluated for their in vitro growth inhibitory effect on A375-C5 (IL-1 insensitive malignant melanoma), MCF-7 (breast adenocarcinoma), and NCI-H460 (non-small-cell lung cancer) cell lines. After promising results were obtained with compound **2**, in contrast to compound **3**, only compound **2** was studied against U251 (glioblastoma astrocytoma), U373 (glioblastoma astrocytoma), and U87MG (glioblastoma astrocytoma) lines. Table 1 shows the concentration that was able to cause 50% cell growth inhibition (GI_50_).

While non-acetylated xanthonoside **3** was not able to reach 50% of cell growth inhibition in the three tested cell lines, the acetylated xanthonoside XGAC (**2**) exhibited a potent growth inhibitory activity in almost all tested cell lines with GI_50_ values between 0.19–0.55 μM, except in IL-1 insensitive A375-C5. These results put forward the role of the acetyl groups in the potent cell growth inhibitory activity of xanthone **2**.

### 2.3. Preparation of Proliposomes and Liposomes for Drug Delivery of Xanthone *(**2**)*

#### 2.3.1. Particle Size, Polidispersity and Zeta Potential

In the hydration studies, the liposomes obtained from the proliposome with xanthone **2** formulations were attained using two types of agitation, specifically, manual and ultrasonic agitation, and two types of solvents, specifically, water or PBS, and their mean diameter can be seen in Table 2. The ultrasonication was able to achieve smaller particles for both solvents when compared to the manual agitation (water, *p* = 0.01; PBS, *p* < 0.01). This difference is probably due to the higher energy input of this technique, which in turn leads to a higher lipid bilayer break during production and smaller spherical structures after self-assembly [38]. The usage of PBS does not improve (ultrasonication, *p* = 0.63) and even increases the mean diameter of particles (manual agitation, *p* < 0.01) (Appendix A). Due to the final users (nurses) when preparing liposomes from proliposomes lack of an ultrasonication apparatus, and since PBS did not improve the sizes, all other tests were made using the manual agitation with water.

The mean diameter, polidispersity (PI) and zeta potential (ZP) of the proliposomes (prepared with manual agitation and water) and liposomes formulations with and without xanthone **2** can be seen in Table 3. The ability of particles to effectively travel to the interstitial space across the tumor vessel walls depends on the ratio particle size/opening size [39,40]. In general, the transport across tumor vessel walls is better when the particles size is smaller [41,42]. The presence of xanthone 2 did not affect the size (*p* > 0.05) for both formulations. The liposomal formulation produced smaller particles, for drug carrying, than the proliposomal formulation (*p* = 0.01), while for the empty formulations the differences were not significant (*p* = 0.51) (Appendix A). As stated before, smaller size liposomes may enhance drug effectiveness. Due to the fast and irregular angiogenesis of tumor tissues, fenestrations and deterioration of blood vessel are common. These are open doors for smaller particles; hence, the smaller the liposome, the higher the possibility that it will leak into the interstitium of the tumor, leading to accumulation, and eventually, to the tumor cell destruction [42].

For all the studied formulations, ZP was more negative than −25 mV, indicating electrostatic stability. For proliposome formulations prepared with manual agitation, the incorporation of xanthone **2** did not change the ZP of liposomes (*p* = 0.132). Finally, for the liposome formulations, the incorporation of xanthone **2** (XGAC) decreased the ZP of liposomes (*p* = 0.022). For all the studied formulations, PI was lower or around 0.3, much lower than 0.7, which is generally indicated as a limit for monodisperse preparations [43].

#### 2.3.2. Thermal Behavior

The differential scanning calorimetry (DSC) thermograms of the mixture of components of the proliposomal formulation is shown in Figure 1A. Nevertheless, in the proliposomal formulation (Figure 1A), only the peak of mannitol (carrier) was visible due to the fact that PC, xanthone **2** and CH are molecularly dispersed in mannitol. The presence of lipids in the formulation decreases the onset temperature of the mannitol peak, revealing the interaction between the lipid part and the carrier material of the formulation [28]. The presence of xanthone **2** in the formulation did not alter the onset temperature of the mannitol peak [28,44]. Figure 1B shows the DSC thermograms of liposomes with and without xanthone **2** and each compound was isolated (CH, PC, xanthone **2**). As it is possible to observe in the thermogram, the lipids and xanthone **2** were well dispersed and did not present crystalline forms. Hence, no peak for the lipids was visible [45,46].

#### 2.3.3. Scanning Electron Microscopic Study

The cryoscanning electron microscopic (Cryo-SEM) technique was used to obtain images of the particles after proliposomes hydration to evaluate their morphology. Figure 2A shows the particles obtained from proliposomes produced by spray drying, without xanthone **2**. The particles observed had a spherical shape. Figure 2B shows the morphology of the particles obtained by hydration of proliposomes produced by SD, with xanthone **2**. As observed, the presence of xanthone **2** in proliposomes did not alter the morphology. Both types of particles presented a very small size, but it is possible to observe particles of larger sizes. Figure 2C,D showed the liposomes, without and with xanthone **2**, respectively. Particles having a spherical shape were observed, and it is possible to conclude that the presence of xanthone **2** in liposomes did not alter their morphology.

The surface morphology of proliposome powders without and with xanthone **2** produced from the SD method was examined by SEM. Figure 3A,B show the spherical particles, with a high surface area. In addition, the incorporation of xanthone **2** within proliposomes did not alter the particles morphology. The amount of proliposomes seen in the conventional SEM was much higher, when compared with Cryo-SEM, since they were not submitted to hydration. The SD seems to produce uniform powdered particles, allowing a fast dispersion of the powder when hydrated to form liposomes.

#### 2.3.4. Entrapment Efficiency

The UV spectrum of xanthone **2** was performed, varying the wavelength from 200 to 400 nm. One of the peaks of xanthone **2** absorption occurred at 300 nm wavelength, which was chosen to detect the compound by the HPLC method for the entrapment efficiency (EE). Data were fitted to the least squares linear regression, and a calibration curve was obtained, with the corresponding equation of the curve y = 0.0665x − 0.6114 having a correlation coefficient (R) of 0.9992, which demonstrates good linearity in the tested range for xanthone **2**.

The EE of xanthone **2** in liposomes obtained by hydration of proliposomes was determined by direct measurement of the compound that was encapsulated in the formulation by an HPLC method. Proliposome formulations have higher drug entrapment efficiency (87.1%) than liposomes (80.2%). Nevertheless, both percentages are relatively high but a significant difference (*p* = 0.049) was found (Appendix A). The high EE is probably due to the lipophilic character of xanthone **2** [47,48]. However, the entrapment efficiency also depends of the lipid composition and of the amount of cholesterol in the formulation [49]. The EE obtained in this work, was close to that obtained with other studies [20,47,50,51].

#### 2.3.5. Stability Study

The obtained proliposomes did not show any visible agglomeration and had the appearance of a free flowing powder. After hydration, proliposomes quickly formed liposome vesicles. This powder showed no apparent changes, after storage for a period of 15, 30, and 90 days indicating the probable stability of the prepared proliposomes.

Proliposomes with and without drug concerning size, ZP and PI, did not present significant statistical differences over time (*p* > 0.05). Fifteen days after, the proliposomes without and with xanthone **2** had a size of 207 ± 38 nm and 196 ± 22 nm, respectively (*p* = 0.685). The ZP values were not statistically significant (*p* = 0.250) and were close to −30 mV. For the proliposomes hydrated after one month, the particles without and with xanthone **2** (XGAC) had a size of 276 ± 7 nm and 226 ± 55 nm, respectively (*p* = 0.191). Concerning ZP, the values were also close to −30 mV, with no statistically significant differences (*p* = 0.142). For the hydration of proliposomes after three months, the particles without and with xanthone **2** had a mean size of 195 ± 64 nm and 177 ± 42 nm (*p* = 0.704), and ZP values close to −30 mV, again with no statistical differences (*p* = 0.249). Liposomes preparations were not stable enough to be measured after the first time interval (Figure 4).

### 2.4. Tumor Cell Viability Effect with Xanthone 2 Formulations

The different formulations of xanthone **2**, liposomes with and without xanthone **2** and proliposomes with and without xanthone **2** were evaluated in glioma cells (U251, U373 e U87MG) to verify their effects on cell viability (Figure 5). The xanthone **2** showed an inhibitory effect in all tested cell lines. For U251, the initiatory effect was achieved for 1 (*p* = 0.031), 10 (*p* < 0.001), and 100 (*p* < 0.001) concentrations. For U373 line, only the 10 (*p* = 0.004) and 100 μM (*p* < 0.001) concentration produced significant values, while for U87MG all concentrations were able to inhibit cell growth, with the exception of 0.1 μM. The empty liposomes showed biocompatible for every line and concentration, except for 100 μM in the cell line U251 (*p* = 0.001) and U373 (*p* = 0.027). While carrying the drug, the liposomal formulation ensured cell inhibition for the cell lines U251 (10 (*p* = 0.022) and 100 μM (*p* < 0.001)) and U373 (1 (*p* = 0.035), 10 (*p* = 0.05) and 100 μM (*p* < 0.001)); yet this effect is lower when compared with the free drug. For the U87MG line, no cell inhibition was achieved using this carrier. In the case of proliposomes, the empty formulation showed to be biocompatible for the cell lines U87MG and U373, except for the 100 μM exposure (*p* = 0.008 e *p* < 0.001, respectively). While for the U251MG line the 1 (*p* < 0.001), 10 (*p* < 0.001) and 100 μM (*p* < 0.001) exposures led to the loss of viability. When a drug is encapsulated, the proliposome formulation showed similar results to the proliposome drug-free formulation, with effect for the cell lines U87MG and U373 in the 100 μM (*p* = 0.008 e *p* < 0.001, respectively), and for U251 in 1 (*p* = 0.012), 10 (*p* < 0.001) e 100 μM (*p* < 0.001) concentrations (Appendix A).

An increase in cell viability of U87MG and U373 after exposure to empty and proliposomes could be noted. This increase is most marked in the case of the empty proliposomes and may be due to mannitol that is present in the formulation. Since mannitol is a sugar, an osmotic effect may occur leading to cell death for the maximum concentration. The same happens for the proliposomes with xanthone **2**, since they present the same formulation. Liposomes clearly showed better results than proliposomes. Future studies have to be performed to correct the osmotic effect of mannitol in the proliposome, to substitute the amount of the carrier in the formulation or to replace mannitol by another transporter (e.g., sorbitol, lactose monohydrate, or other lipids in the formulation) [52,53].

## 3. Materials and Methods

### 3.1. General Information

Reagents were mainly purchased from Sigma-Aldrich Co. (Sintra, Portugal). Lipoid E80 (egg phospholipids with 80% of phosphatidylcholine) was acquired from Lipoid, cholesterol from Acofarma and methanol (HPLC grade) from VWR chemicals. Reactions were controlled by thin layer chromatography (TLC) using Merck silica gel 60 (GF254) plates. Compounds were visually detected by absorbance at 254 and/or 365 nm. IR spectra were obtained in KBr microplate in a FTIR spectrometer Nicolet is 10 from Thermo Scientific with Smart OMNI-Transmission accessory (Software OMNIC 8.3, Thermo Scientific, Madison, WI, USA). ^1^H and ^13^C-NMR spectra were performed in the Department of Chemistry from the University of Aveiro, Portugal, on a Bruker Avance 300 instrument (^1^H: 300.13 MHz; ^13^C: 75.47). ^13^C-NMR assignments were made by bidimensional HSQC and HMBC experiments (long-range C, H coupling constants were optimized to 7 and 1 Hz). Chemical shifts are expressed in ppm values relative to tetramethylsilane (TMS) as an internal reference and coupling constants are reported in hertz (Hz). HRMS mass spectra were performed in Centro de Apoio Científico e Tecnolóxico á Investigation (CACTI) from the University of Vigo, Spain, on an APEX III mass spectrometer, in ESI (Electrospray) mode. Six human tumor cell lines were used in this study: A375-C5 (IL-1 insensitive), MCF-7 ER (+), NCI-H460, U251, U373, and U87MG.

### 3.2. Synthesis of 3,6-Bis(2,3,4,6-tetra-O-acetyl-β-glucopyranosyl)xanthone (**2**, XGAC)

*Michael reaction method*: 3,6-Dihydroxyxanthone (**1**, 0.800 g, 3.5 mmol) was dissolved in water (100 mL) with potassium carbonate (9.6 g, 70 mmol; 10 equiv/OH). After adding 100 mL of chloroform, 2,3,4,6-tetra-*O*-acetyl-α-d-glucopyranosyl bromide (12 g, 35 mmol; 5 equiv/OH) and tetrabuthylammonium bromide (16 g; 52.5 mmol; 7.5 equiv/OH) were added and the biphasic reaction was left with vigorous stirring at room temperature. After 90 h the potassium carbonate was filtered out. The filtrate was concentrated, alkalinized with aqueous sodium hydroxide and extracted with chloroform (3 × 30 mL). The joined organic phases were dried over sodium sulfate and the solvent removed under reduced pressure. A white solid, insolubilized from the crude product when cold methanol, was added (**2**; 20% yield).

Koenigs-Knorr method: To a stirred solution of 3,6-dihydroxyxanthone (**1**, 3 g; 13.5 mmol) and 2,3,4,6-tetra-*O*-acetyl-α-d-glucopyranosyl bromide (22.10 g; 54.40 mmol; 2 equiv/OH) in acetonitrile/acetone, silver carbonate (11.20 g; 40.6 mmol; 1.5 equiv/OH) was carefully added in portions at room temperature. The mixture was stirred in the dark for 48 h, and then filtered under reduced pressure. The filtrate was purified by flash column chromatography (100% CHCl_3_) (**2**, XGAC; 7% yield).

Melting point: 255 °C–257 °C; IR (KBr) νmax: 1754, 1613, 1497, 1443, 1377,1315, 1223, 1181, 1036, 986, 929, 905, 859, 771, 670, 599 cm^−1^; ^1^H-NMR (CDCl_3_, 300.13 MHz) δ: 8.27 (1H, d, *J* = 9.4 Hz, H-1/H-8), 7.03 (1H, d, *J* = 2.2 Hz, H-4/H-5), 7.00–6.99 (1H, m, H-2/H-7), 5.36–5.17 (4H, m), 4.31 (1H, dd, H-6′a, *J* = 12.3 Hz, *J* = 5.6 Hz) 4.22 (1H, dd, H-6′b, *J* = 12.3 Hz, *J* = 2.4 Hz), 3.97 (1H, m, H-5′), 2.12 (3H, s), 2.09 (3H, s, COC*H_3_*), 2.08 (6H, s, COC*H_3_*), 2.06 (3H, s, COC*H_3_*); ^13^C-NMR (CDCl_3_, 75.47 MHz) δ: 175.3 (C-9), 170.5 (*C*OCH_3_), 170.2 (*C*OCH_3_), 169.4 (*C*OCH_3_), 169.3 (*C*OCH_3_), 161.4 (C-3, C-6), 157.5 (C-4a, C-10a), 128.7 (C-1, C-8), 117.5 (C-8a, C-9a), 113.8 (C-2, C-7), 104.1 (C-4, C-5), 98.2 (C-1′), 72.5 (C-5′), 72.4 (C-3′), 70.9 (C-2′), 68.1 (C-4′), 61.9 (C-6′), 20.7 (CO*C*H_3_), 20.6 (CO*C*H_3_), and 20.6 (2 CO*C*H_3_). HRMS (ESI^+^) *m*/*z* calcd for C_41_H_45_O_22_ [M + H]^+^ 889.23828, found 889.23970.

### 3.3. Cell Lines and Culture Conditions

The human tumor cell lines used were A375-C5 (melanoma), MCF-7 (breast adenocarcinoma), and NCI-H460 (non-small cell lung cancer), U251 (glioblastoma astrocytoma), U373 (glioblastoma astrocytoma), and U87MG (glioblastoma astrocytoma). A374-C5, MCF-7, and NCI-H460 were grown in RPMI-1640 (Biochrom, Berlin, Germany), and U251, U373, and U87MG in DMEM (Biochrom). Media were supplemented with 5% heat-inactivated fetal bovine serum (FBS, Biochrom). Cell lines were grown at 37 °C in a 5% CO_2_ humidified atmosphere (Hera Cell, Heraeus), and cell viability was monitored by Trypan Blue exclusion assay to ensure at least 95% viability before use.

### 3.4. Tumor Cell Growth Assay

The antigrowth activity of the compounds was assessed by the sulforhodamine B (SRB) assay, as adopted from the National Cancer Institute (NCI) in the ‘In vitro Anticancer Drug Discovery Screen. Cells were plated in 96-well plates at 0.05 × 106 cells/100 μL/well in complete medium at 37 °C for 24 h. Then, cells two-fold serial dilutions of the test compounds were added to culture medium, at concentrations ranging from 0 to 150 µM, for 48 h. Control groups were treated with equivalent amount, up to 0.25% concentration, of the solvent dimethyl sulfoxide (DMSO, Sigma-Aldrich). Then, cells were fixed with 50% (*m*/*v*) trichloroacetic acid (Merck Millipore, Darmstadt, Germany), washed with distilled water and stained with Sulforhodamine B (SRB; Sigma-Aldrich) for 30 min at room temperature. After a 5 times wash with 1% (*v*/*v*) acetic acid (Merck Millipore), plates were left to dry at room temperature before SRB complex solubilization with 10 mM Tris buffer (Sigma-Aldrich) for 30 min. Cell survival was measured through determination of the absorbance at 515 nm in a microplate reader (Biotek Synergy 2,BioTek Instruments, Inc., Winooski, VT, USA). The 50% cell growth inhibition concentration (GI_50_) was calculated using the dose–response curve established for each test compound.

### 3.5. Preparation of Proliposomes

Choosing the best mass ratio egg phosphatidylcholine:cholesterol (3:1) and lipids:mannitol (1:10) previously obtained [25,26], proliposomes with and without xanthone **2** were produced using spray-drying (SD) method, with 1% of the drug tested (xanthone **2**) in relation to the mass of lipids. The egg phosphatidylcholine (PC) and cholesterol (CH) were dissolved in ethanol, and the mannitol was dissolved in water; both substances were transferred to a 100 mL volumetric flask. In the preparation of the formulations with xanthone **2**, the drug was dissolved in absolute ethanol along with the lipids. The proliposome formulation was prepared with the SD technique using the Nano Spray Dryer B-90 (Buchi Labortechnik, Switzerland) the experimental conditions used were as follows: Temperature: 80 °C, compressed air flow: 90 L/min, pressure: 30 mbar, spray nozzle membrane: 5.5 μm. After the process was completed, the apparatus was dismantled to remove the inner cylinder, where the particles were collected with the aid of a scraper. At the end of the process, the proliposomes were collected into glass vials and stored in a desiccator [54]. Proliposomes were prepared with and without xanthone **2**.

### 3.6. Preparation of Liposomes

For the classic thin-film hydration technique, the lipids, cholesterol, and egg phosphatidylcholine were mixed in a chloroform:methanol mixture in a 3:1 ratio, according to a well described protocol elsewhere [55,56]. Briefly, in a round-bottom flask, the preparation was dried using a rotary evaporator (R-210 Buchi, Montreal, QC, Canada) for at least 30 min, in a thermostatically controlled water bath at 30 °C under a stream of nitrogen. Hydration of the films was done with the addition of phosphate buffer solution (PBS), and after vortexed to promote the self-assembly of the liposomes. Finally, the liposomes were extruded (Lipex, Burnaby, BC, Canada), three times through 600 nm and 200 nm filters, and then 10 times through a 100 nm filter. 

### 3.7. Characterization of Proliposomes and Liposomes

#### 3.7.1. Hydration Study

Using the prolipossomes powder, the liposomes vesicles were quickly formed by hydration. The process was subjected to dissolution of the carrier and self-assembling of vesicles due hydrophobic repulsion of the apolar chain of the lipids. The proliposome powders were hydrated using either purified water or phosphate buffer solution (PBS) pH = 7.4, mixed by hand and or by sonication to obtain liposomal dispersions. Afterwards, the suspensions were filtered through a 5 μm filter, in order to separate possible aggregates [57]. The effective diameters and PI were evaluated by dynamic light scattering (DLS) [58], using a ZetaPALS apparatus (Brookhaven Instruments Corporation, Holtsville, NY, USA). The data collected by the PALS Particle Sizing Software (Version 5, Brookhaven Instruments Corporation, Holtsville, NY, USA) was expressed as mean ± standard deviation.

The ZP were evaluated by electrophoretic light scattering (ELS) using a ZetaPALS apparatus (Brookhaven Instruments Corporation, Holtsville, NY, USA). The results analyzed by the software PALS Zeta Potential Analyzer (Version 5, Brookhaven Instruments Corporation, Holtsville, NY, USA) were expressed as mean ± standard deviation [59].

#### 3.7.2. Thermal Behavior

The study of the drug–excipient and excipient-excipient compatibility studies were performed using a DSC 200 F3 Maia (Netzsh–Gerätebau GmbH, Germany). Drug, excipients and formulations of proliposomes and liposomes, were weighed directly in the DSC aluminum pans and scanned in a range of temperatures of −40 to 340 °C under a nitrogen atmosphere with a 40 mL/min flow. A heating rate of 10 °C/min was used, and the thermograms obtained were observed for any interaction. An empty aluminum pan was used as a reference. The onset temperatures were calculated using Proteus Analysis software (Version 6.1, Netzsh-Gerätebau GmbH, Germany). The DSC cell was calibrated (sensitivity and temperature calibration) with Hg (m.p. −38.8 °C), In (m.p. 156.6 °C), Sn (m.p. 231.9 °C), Bi (m.p. 271.4 °C), Zn (m.p. 419.5 °C), and CsCl (m.p. 476.0 °C) as standards.

#### 3.7.3. Scanning Electron Microscopic Study

Image analyses of the liposomes and the proliposome powder, loaded and unloaded, was performed using, respectively, Cryo-SEM (JEOL JSM 6301F/Oxford INCA Energy 350/Gatan Alto 2500) and SEM (FEI Quanta 400FEG ESEM/EDAX Genesis X4M) techniques. The Cryo-SEM/EDS samples were rapidly frozen by submerging the formulation in liquid nitrogen (slushy from) and then transferred to the cold vacuum preparation station. Afterward, the frozen samples were fractured (etched) for 90 s at negative 90 °C and coated with a thin film of Au/Pd by sputtering for 45 s (SPI-Module™ Sputter Coater, Structure Probe, Inc; West Chester, PA, USA). Finally, the samples are analyzed using SEM operating at negative 150 °C. For the dry proliposomes powder, the samples were merely coated with Au/Pd using, the same method as before, and transferred to the SEM.

#### 3.7.4. Entrapment Efficiency

Proliposomes were weighed (200 mg) and diluted in 2 mL of ultra-pure water. Diluted samples were filtered through a 5 μm nitrocellulose membrane filter to reject unincorporated xanthone **2**. Subsequently, methanol dilution of 1:8 and manual shaking was performed to allow the release of the xanthone **2,** which was encapsulated in the liposomes. The obtained mixture was centrifuged (5000 rpm for 15 min) (Model 5804, Eppendorf, Hauppauge, NY, USA) and the supernatant filtered (0.45 μm PTFE filter, OlimPeak^®^, Teknokroma, Barcelona, Spain) to obtain a xanthone **2** solution. The obtained samples were then evaluated by HPLC, and each sample was injected three times.

HPLC analysis (Dionex Ultimate 3000 (Thermo Scientific, Darmstadt, Germany) of the samples was performed under the same xanthone **2** assay conditions. By interpolation of the calibration curve, the actual xanthone **2** concentration was obtained. The theoretical concentration of xanthone **2** was calculated by taking into account the amount of xanthone **2** placed during the production of the proliposomes and the dilutions performed throughout the procedure. Thus, the encapsulation efficiency (EE) was calculated as follows:EE(%) = (C_filtrate_/C_formulation_) × 100(1)
where C_filtrate_ is the concentration of the drug in the filtrate (µg/mL) and C_formulation_ is the initial concentration of the drug in the formulation (µg/mL).

#### 3.7.5. Stability Study

The stability study was performed for the liposomal and proliposomal formulations with and without xanthone **2**. Samples were periodically evaluated regarding size, ZP, and PI at time 0 and after 15, 30, and 90 days after production. At defined intervals, the proliposome powder was reconstituted to produce liposomes.

#### 3.7.6. MTT Cell Viability Assay

Cell viability was determined by the MTT (3-(4,5-dimethylthiazolyl-2)-2,5-diphenyltetrazolium bromide) (Sigma-Aldrich) assay, in glioblastoma cell lines. A total of 5 × 10^4^ cells were seeded in 96-well plate and incubated at 37 °C/5% CO_2_. After 24 h, cells were exposed during 48 h to xanthone **2** (XGAC), proliposomes with and without xanthone **2**, and liposomes with and without xanthone **2**. Then, fresh FBS-free medium and 20 μL MTT reagent (5 mg/mL in PBS) were added and cells incubated at 37 °C/5% CO_2_. After 4h incubation, a detergent solution (89% *v/v*) 2-propanol, 10% (*v/v*) Triton X-100, 1% (*v/v*) HCl 3.7%(*v/v*) was added for 2 h to solubilize the purple formazan crystals. Cell survival was deduced through determination of the absorbance at 570 nm in a microplate reader (Biotek Synergy 2,BioTek Instruments, Inc., Winooski, VT, USA).

### 3.8. Statistical Analysis

The results are shown as the mean ± standard deviation of three batches of the same formulation. The results of ZP, mean diameter, EE, and cell viability were statistically analyzed using ANOVA or t-student test, after confirming the normality and homogeneity of the variance with the Shapiro-Wilk and Levene tests. Differences between groups for ANOVA were compared with a post hoc test (Tukey HSD or Dunnett). Significance was set at *p* < 0.05 or at *p* < 0.01. All the statistical analyzes were performed with the software, IBM SPSS Statistics for Windows (Version 25.0.: IBM Co., Armonk, NY, USA).

## 4. Conclusions

In this work, a potent antitumor synthetic xanthone derivative (compound **2**) was synthesized and proliposomes and liposomes have been developed as drug carriers. This acetylated xanthonoside presents poor water solubility and might suffer rapid hydrolysis by esterases being its encapsulation in nanosystems a possible strategy to overcome these limitations. The results showed that the liposomes obtained from the proliposomes have an average diameter of 200 nm. In comparison, the traditional liposomes presented an average size in the order of 100 nm. Both formulations presented negative zeta potential values. The stability of the proliposomes was tested and showed no significant changes in liposome properties after 15, 30, and 90 days. The use of the Cryo-SEM and SEM techniques allowed the evaluation of the morphology of the proliposomes and liposomes formulations, presenting both as spherical particles with uniform morphology. The drug-free proliposomal formulation presented some toxicity, showing that new proliposomal formulations with different mannitol concentrations or different carriers will be needed. However, xanthone **2** was successfully incorporated in liposomes maintaining the inhibitory activity against glioblastoma cell lines. This result highlighted that it will be worthy to explore new formulations and/or drug carriers to fight glioma growth with this promising xanthone.

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
