# Peer review of "Discovery of a New Xanthone against Glioma: Synthesis and Development of (Pro)liposome Formulations"

_molecules, 2019, doi:10.3390/molecules24030409_

Round 1
Reviewer 1 Report
The manuscript describes a novel modified xanthone, it’s incorporation into pro liposomes, and use on cancer cell lines. The work is novel, the techniques largely suitable for the goals, there is potential for this work to have an impact even though the cell growth inhibition numbers are modest.
I recommend the manuscript be revised to clarify some of the techniques used.
Evidence that the beta anomer of 2 and 3 were obtained is not discussed in the manuscript. The authors should briefly mention the proton or carbon assignments for the anomeric positions in the manuscript. The HMBC and HSQC spectra could also be included in supporting info.
The data discussed in section 2.3.4 on entrapment efficiency should be included as a figure, possibly in supporting information.
The discussion of table 2 includes a test of significance for only one of the possible comparisons, liposomes to liposomes with 2. The same test should also be reported for the other two preparations.
The description of the liposome preparation method is concerning. The materials section reads as though a rotovap was used, followed by nitrogen. That is reversed compared to what is typical. Also 30 min of rotovap is not sufficient to remove methanol from lipids under typical conditions. High vacuum and longer times are more commonly required.
The lack of a peak for PC proliposomes and liposomes in the thermograms in figure 1 is not adequately explained. Why does the PC data stop at 120 C? And what PC was used? Saturated or unsaturated, and what chain length? Also, X2 is an unfortunate abbreviation as that is commonly used to indicate a doubling the y-axis for a sample.
Author Response
Reviewer 1
The manuscript describes a novel modified xanthone, it’s incorporation into pro liposomes, and use on cancer cell lines. The work is novel, the techniques largely suitable for the goals, there is potential for this work to have an impact even though the cell growth inhibition numbers are modest.
I recommend the manuscript be revised to clarify some of the techniques used.
1. Evidence that the beta anomer of 2 and 3 were obtained is not discussed in the manuscript. The authors should briefly mention the proton or carbon assignments for the anomeric positions in the manuscript. The HMBC and HSQC spectra could also be included in supporting info.
Reply: We thank the reviewer comment. We added the missing discussion of the anomeric configuration: “The inversion of stereochemistry at the anomeric center of the glucose donor was confirmed by the coupling constant of 7 Hz of the doublet at δH 5.24 ppm (H-1’), indicating a β-configuration.” The HMBC and HSQC spectra were also included in supporting info (Figure S1).
2. The data discussed in section 2.3.4 on entrapment efficiency should be included as a figure, possibly in supporting information.
Reply: We added a figure and a table with the statistical results in the Supplementary Materials (Figure S3 and Table S3).
3. The discussion of table 2 includes a test of significance for only one of the possible comparisons, liposomes to liposomes with 2. The same test should also be reported for the other two preparations.
Reply: We added a new Table 2 (Mean diameter for formulations of proliposome with xanthone 2 obtained by spray drying (SD), prepared by different methods) and the influence of agitation method and solvent was discussed. In Table 3 (Results obtained (mean diameter, PI and ZP) for proliposome (prepared with manual agitation and water) and liposomal formulations, with and without xanthone 2 (XGAC)) we discuss the type of particle and the influence of the drug. The statistical results were added to the Supplementary Materials (Table S1 and Table S2).
4. The description of the liposome preparation method is concerning. The materials section reads as though a rotovap was used, followed by nitrogen. That is reversed compared to what is typical. Also 30 min of rotovap is not sufficient to remove methanol from lipids under typical conditions. High vacuum and longer times are more commonly required.
Reply: We understand the Reviewer’s concerns. However, the described method is widely used both by our group and by many others [1,2,3]. Additionally, it is also the standard procedure from the manufacturer of the lipids (https://www.sigmaaldrich.com/technical-documents/articles/biology/liposome-preparation.html). Moreover, and as stated in the manuscript, the solvent mixture is evaporated in a rotatory evaporator under a stream of nitrogen, and not followed by nitrogen. The nitrogen flux aids the evaporation of the solvents while prevents oxidation. Furthermore, and we will correct this on the manuscript, the evaporation was at least 30 minutes. The time was depended on the volume of the solution, which typically ranged from 2 up to 5 mL at the most. For these volumes, in particular 2 mL, 30 minutes, at 30 °C under a flux of nitrogen at 3 bar is enough to ensure solvent evaporation.
Refs:
[1] H. Chakraborty, M. Sarkar, Interaction of piroxicam and meloxicam with DMPG/DMPC mixed vesicles: anomalous partitioning behavior, Biophys Chem 125(2-3) (2007) 306-13.
[2] C. Nunes, G. Brezesinski, C. Pereira-Leite, J.L.F.C. Lima, S. Reis, M. Lucio, NSAIDs Interactions with Membranes: A Biophysical Approach, Langmuir 27(17) (2011) 10847-10858.
[3] D.D. Lasic, Liposomes - from Physics to Applications, Elsevier, New York, 1993.
5. The lack of a peak for PC proliposomes and liposomes in the thermograms in figure 1 is not adequately explained. Why does the PC data stop at 120 C? And what PC was used? Saturated or unsaturated, and what chain length?
Reply: PC= Lipoid E80 (egg phospholipids with 80% of phosphatidylcholine) was acquired from Lipoid – contain ca. 20% w/w unsatured PE. The used PC, is a natural mixture extracted from egg yolk, thus presents several chain lengths as well as unsaturated and saturated chains. This mixture of different carbon chain lengths and saturations, leads to a range of temperatures of phase transition and not a specific one, thereby explaining the very broad and almost negligible band that we have around 20 °C. Regarding the proliposomes, the phase transition of mannitol is so well defined that covers the broad band regarding to PC. To clarify this point we add a thermogram for PC until 350 °C. The PC, XGAC and CH components are molecularly dispersed in mannitol and so they do not present a melting peak.
Also, X2 is an unfortunate abbreviation as that is commonly used to indicate a doubling the y-axis for a sample.
Reply: The abbreviation X2 in the Figures 1, 4 and 5 was changed to XGAC.
Reviewer 2 Report
Alves et. al describe the encapsulation of xanthone in liposomes and proliposomes for potential use in glioma therapy. The potential advance is the encapsulation of xanthones in proliposomes for long term stability, but the data suggest one of the following: (1) toxicity of the proliposome formulation or (2) a combination of blank proliposome toxicity and a lack of xanthone release from the proliposomes. Because the preparation of xanthone liposomes have already been reported, it is unclear what this work contributes to the field. Optimization of the proliposome formulation before resubmission is recommended for the work to be of higher impact.
1. A xanthone liposome has been reported by Chen et. al. in Nanomedicine: Nanotechnology, Biology, and Medicine in 2016, which should be cited. In addition, the liposome in this work is transferrin-modified, which enhances targeting. How are the liposomes/proliposomes in this work potentially better than the liposome in this work?
2. What is the amount of xanthone release from liposomes vs. proliposomes over time?
3. For better presentation, please present the size/zeta potential data in the stability study as a plot over time.
4. Two synthesis methods – manual agitation and ultrasonication – were used to formulate proliposomes. Which synthesis method for the proliposomes was used in the cell culture study? The synthesis method affects the size. Is entrapment efficiency or drug release affected as well? A comparison of efficacy between liposomes of different synthesis methods could provide interesting insights. With liposome vs. manually agitated proliposome vs. ultrasonicated proliposome – maybe carrier-based cytotoxicity is size-driven.
5. Mannitol is discussed as a potential cause of cytotoxicity for cell death – have the authors tried different excipients for proliposome preparation?
6. In the methods, it is stated that six human tumor cells were used to study antitumor activity, but data from only three tumor lines are shown. The data from the other three cell lines needs to be shown or these cell lines should be omitted from the methods.
7. Statistical differences between groups should be shown in the cell viability experiments. Statistical analysis was mentioned in the methods but not shown in the data.
Author Response
Reviewer 2
Alves et. al describe the encapsulation of xanthone in liposomes and proliposomes for potential use in glioma therapy. The potential advance is the encapsulation of xanthones in proliposomes for long term stability, but the data suggest one of the following: (1) toxicity of the proliposome formulation or (2) a combination of blank proliposome toxicity and a lack of xanthone release from the proliposomes. Because the preparation of xanthone liposomes have already been reported, it is unclear what this work contributes to the field. Optimization of the proliposome formulation before resubmission is recommended for the work to be of higher impact.
1. A xanthone liposome has been reported by Chen et. al. in Nanomedicine: Nanotechnology, Biology, and Medicine in 2016, which should be cited. In addition, the liposome in this work is transferrin-modified, which enhances targeting. How are the liposomes/proliposomes in this work potentially better than the liposome in this work?
Reply: We agree with the reviewer comment. We add the reference missing and rewrite/rediscuss the potential advantages of encapsulation of xanthones.
Both the therapeutic objective as the drug used here are different and this one showed anti-cancer activity and so the text was changed to reinforce this idea. This xanthone is insoluble in water by which cannot be administered directly in solution. Furthermore, because the liposomes formulations have low physical stability with a low shelf life, usually less than a week, a formulation of proliposomes was studied that showed higher stability since it can be held in the dry state and prepared the liposomes extemporaneously.
2. What is the amount of xanthone release from liposomes vs. proliposomes over time?
Reply: Since the purpose is that the integral liposome suffers endocytosis by the cells of glioma, all the xanthone X2 contents will be absorbed, not just the amount released, and that’s why we did not study the release over time. The liposomes will be better absorbed due to the weakening of the BBB near the glioma (passive targeted based on the EPR effect) as referred in [1]. The text was changed to reinforce this idea.
[1] Kim, S. S.; Harford, J. B.; Pirollo, K. F.; Chang, E. H., Effective treatment of glioblastoma requires crossing the blood-brain barrier and targeting tumors including cancer stem cells: The promise of nanomedicine. Biochem Biophys Res Commun 2015, 468, (3), 485-9.
3. For better presentation, please present the size/zeta potential data in the stability study as a plot over time.
Reply: A new figure containing the size, ZP and PI was added to the work showing the stability over time (Figure 4).
4. Two synthesis methods – manual agitation and ultrasonication – were used to formulate proliposomes. Which synthesis method for the proliposomes was used in the cell culture study? The synthesis method affects the size. Is entrapment efficiency or drug release affected as well? A comparison of efficacy between liposomes of different synthesis methods could provide interesting insights. With liposome vs. manually agitated proliposome vs. ultrasonicated proliposome – maybe carrier-based cytotoxicity is size-driven.
Reply: Because final users (nurses) when preparing liposomes from proliposomes lack of a ultrasonication apparatus all other tests were made using the manual agitation. This issue was clarified in the work.
5. Mannitol is discussed as a potential cause of cytotoxicity for cell death – have the authors tried different excipients for proliposome preparation?
Reply: Mannitol was chosen as the carrier because most works published for the preparation of proliposomes, their formulations referred it as the carrier [2-5]. To our knowledge, no proliposome formulation using mannitol was ever tested for biocompability, enhancing the novelty of this work in relation to practical use of mannitol.
2. Omer, H. K.; Hussein, N. R.; Ferraz, A.; Najlah, M.; Ahmed, W.; Taylor, K. M. G.; Elhissi, A. M. A., Spray-Dried Proliposome Microparticles for High-Performance Aerosol Delivery Using a Monodose Powder Inhaler. AAPS PharmSciTech 2018.
3. Ye, T.; Sun, S.; Sugianto, T. D.; Tang, P.; Parumasivam, T.; Chang, Y. K.; Astudillo, A.; Wang, S.; Chan, H.-K., Novel combination proliposomes containing tobramycin and clarithromycin effective against Pseudomonas aeruginosa biofilms. International Journal of Pharmaceutics 2018, 552, (1), 130-138.
4. Ahammed, V.; Narayan, R.; Paul, J.; Nayak, Y.; Roy, B.; Shavi, G. V.; Nayak, U. Y., Development and in vivo evaluation of functionalized ritonavir proliposomes for lymphatic targeting. Life Sci 2017, 183, 11-20.
5. Hao, F.; He, Y.; Sun, Y.; Zheng, B.; Liu, Y.; Wang, X.; Zhang, Y.; Lee, R. J.; Teng, L.; Xie, J., Improvement of oral availability of ginseng fruit saponins by a proliposome delivery system containing sodium deoxycholate. Saudi Journal of Biological Sciences 2016, 23, (1, Supplement), S113-S125.
6. In the methods, it is stated that six human tumor cells were used to study antitumor activity, but data from only three tumor lines are shown. The data from the other three cell lines needs to be shown or these cell lines should be omitted from the methods.
Reply: We thank the reviewer for this comment. Since compound 2 was very potent in 2 of the 3 cell lines routinely tested, we judged interesting to test its activity in glioma cells also as very recently in our group (Neves et al. Molecules 2018) a related compound, an acetylated flavonoid glycoside, was shown to have a potent inhibitory effect on the growth of glioma cell lines.
7. Statistical differences between groups should be shown in the cell viability experiments. Statistical analysis was mentioned in the methods but not shown in the data.
Reply: Please note that statistical differences between groups were described in the text of the 2.4. section of the original manuscript. We now added these statistics to figure 5 to make reading easier in the revised manuscript. We also added the full statistical results (Multiple Comparison) to the Supplementary Materials (Table S4).
Reviewer 3 Report
Major comments:
1. Cell growth inhibitory activity for compound 1 is not shown in Table 1. Please add this result in Table 1 and discuss it when compare with compounds 2 and 3. At least, the authors should be mentioned why they select the compound 2 rather than compound 1.
2. The statistics for Fig. 4 is not available. Please add the statistics marks and description.
3. The drug treatment time for cell viability determination is not described in Material section or result (Table footnote). Please add this information.
4. What is the method for Table 1 and Fig. 4? In Material section, SRB (3.3) assay and MTT (3.6.6) assay are mentioned. But it is not pointed out what method is applied to which one.
Minor comments:
1. Table 1. Please add the footnote with data = mean +- SD (N = ?)
2. I cannot understand the Y-axis for Fig. 4. What is the “mean” for? Please describe in detail and use more suitable unit.
3. What is the unit for PI in Table 2? Data = mean +_ SD? ( n= ?)
Author Response
Reviewer 3
Major comments:
1. Cell growth inhibitory activity for compound 1 is not shown in Table 1. Please add this result in Table 1 and discuss it when compare with compounds 2 and 3. At least, the authors should be mentioned why they select the compound 2 rather than compound 1.
Reply: We understand the reviewer concern and agree with this comment. We had previously tested compound 1 in a panel of 3 tumor cell lines (Costa et al. Letters in Drug Design & Discovery 2010) including glioma and it was not very active (MCF-7 R(+) (breast) NCI H460 (lung), SF-268 (glioma) with GI50 values of, 74.6, 31.1, and 149.1 uM respectively). The purpose of the work presented in this manuscript was the investigation of the xanthone glycoside scaffold with expectations of obtaining derivatives active against glioma based on our recent results with flavonoids (Neves et al. Molecules 2018). Nevertheless, in the revised manuscript to a better understanding of the rational of this work we introduce the information of compound 1 and rediscuss the results.
2. The statistics for Fig. 4 is not available. Please add the statistics marks and description.
Reply: The figure was supplemented with the statistical information. We also added the full statistical results (Multiple Comparison) to the Supplementary Materials (Table S4).
3. The drug treatment time for cell viability determination is not described in Material section or result (Table footnote). Please add this information.
Reply: We apologize for this missing information. To include this information, we now altered the Table footnote to “Results are presented as the concentrations that cause 50% cell growth inhibition (GI50) after a continuous exposure of 48 hours, and represent means ± SEM from at least three independent experiments, as determined by SRB. Doxorubicin was used as positive control. Nd – not determined.”
4. What is the method for Table 1 and Fig. 4? In Material section, SRB (3.3) assay and MTT (3.6.6) assay are mentioned. But it is not pointed out what method is applied to which one.
Reply: We performed SRB for results in Table 1 and MTT for results in Fig. 4. We now added this missing information to Table 1 footnote (please see our answer to Reviewer 3 comment) and to Fig. 5 legend, respectively.
Minor comments:
1. Table 1. Please add the footnote with data = mean +- SD (N = ?)
Reply: Done.
2. I cannot understand the Y-axis for Fig. 4. What is the “mean” for? Please describe in detail and use more suitable unit.
Reply: Corrected.
3. What is the unit for PI in Table 2? Data = mean +_ SD? (n= 3)
Reply: Corrected (as referred in 3.7. Statistical analysis).
Round 2
Reviewer 2 Report
The authors have addressed my comments satisfactorily.